# Phosphorylation and Dephosphorylation of Beta-Amyloid Peptide in Model Cell Cultures: The Role of Cellular Protein Kinases and Phosphatases

**DOI:** 10.3390/life13010147

**Published:** 2023-01-04

**Authors:** Evgeny P. Barykin, Dmitry V. Yanvarev, Maria A. Strelkova, Vladimir T. Valuev-Elliston, Kseniya B. Varshavskaya, Vladimir A. Mitkevich, Alexander A. Makarov

**Affiliations:** Engelhardt Institute of Molecular Biology, Vavilov Str. 32, 119991 Moscow, Russia

**Keywords:** beta-amyloid, phosphorylation, casein kinase 2, blood–brain barrier, radiolabelling, phosphatase

## Abstract

Phosphorylation of beta-amyloid peptide (Aβ) at the Ser8 residue affects its neurotoxicity, metal-dependent oligomerisation, amyloidogenicity, and other pathogenic properties. Phosphorylated Aβ (pS8-Aβ) was detected in vivo in AD model mice and in the brains of patients with AD. However, the pS8-Aβ production and the regulation of its levels have not been previously studied in detail. In this paper, immunochemical methods together with radioactive labelling were used to study the Aβ phosphorylation by intracellular and surface protein kinases of HEK293 cells and brain endothelial cells (bEnd.3). It was found that HEK293 robustly phosphorylated Aβ, likely with contribution from casein kinase 2 (CK2), whereas in bEnd.3, the activity of Aβ phosphorylation was relatively low. Further, the study showed that both HEK293 and bEnd.3 could dephosphorylate pS8-Aβ, mainly due to the activity of protein phosphatases PP1 and PP2A. The Aβ dephosphorylation efficiency in bEnd.3 was three times higher than in HEK293, which correlated with the reduced abundance of pS8-Aβ in vascular amyloid deposits of patients with AD compared to senile plaques. These data suggest an important role of CK2, PP1, and PP2A as regulators of Aβ phosphorylation, and point to the involvement of the blood–brain barrier in the control of Aβ modification levels.

## 1. Introduction

Alzheimer’s disease (AD) is the most common form of dementia, resulting from neurodegenerative changes in the brain. The progression of AD represents a pathological cascade most likely starting with the accumulation and aggregation of beta-amyloid peptide (Aβ) [1,2]. Aβ is produced by the proteolytic cleavage of APP protein, which occurs both at the outer side of the plasma membrane and inside the cell—in the Golgi apparatus and vesicles [3]. Aβ undergoes various post-translational modifications that affect its properties and can either provoke its aggregation and increase neurotoxicity or prevent the manifestation of its pathogenic properties [4,5]. Changes in the levels of different Aβ modifications are associated with AD [6,7], and the study of the role of the modifications is necessary to understand the pathogenesis of the disease and develop new biomarkers and therapeutic approaches.

Phosphorylation of Aβ at Ser8 is a modification found in the brains of patients with AD and Down syndrome [7,8]. In the brains of mice of model lines, early intracellular accumulation of phosphorylated Aβ (pS8-Aβ) is observed, which precedes the formation of plaques, and pS8-Aβ has increased direct neurotoxicity in vitro compared to the non-phosphorylated peptide [7,9]. On the other hand, subsequent studies have shown that Aβ phosphorylation prevents its aggregation in the presence of zinc ions, and injections of pS8-Aβ slow down cerebral amyloidogenesis in mouse models of AD [10]. It has been shown that Aβ can be modified in vitro by neuronal surface kinases and by soluble kinases of the cerebrospinal fluid [11]. The properties of pS8-Aβ make it an important form of the peptide for the pathogenesis of AD; however, the origin of pS8-Aβ in the body and the regulation of its levels have not been studied enough. It is not known whether the formation of pS8-Aβ occurs in the blood, in the brain tissue, or whether its modification occurs during the transport of Aβ across the blood–brain barrier (BBB). When Aβ becomes internalised in cells, it disrupts the functioning of proteasomes and mitochondria [12,13]; however, the Aβ modification by intracellular enzymes remains undetermined. The level of phosphoproteins in the body is associated not only with the activity of kinases, but also with the intensity of dephosphorylation [14]. The possibility of Aβ dephosphorylation by different cells and the key enzymes involved in this process are also unknown. In this work, the research team assessed the phosphorylation of 42 a.a. long Aβ isoform (Aβ) under various conditions—inside or on the surface of mouse brain endothelial cells bEnd.3 and the HEK293 cell line—and also probed the possibility of the dephosphorylation of pS8-Aβ by cell phosphatases.

## 2. Materials and Methods

### 2.1. Preparation of Beta-Amyloid Peptides

Synthetic beta-amyloid peptide Aβ_42_:

[H2N]-DAEFRHDSGYEVHHQKLVFFAEDVGSNKGAIIGLMVGGVVIA-[COOH] (Aβ_42_) was purchased from Lifetein (Somerset, NJ, USA). Aβ_42_ phosphorylated at Ser8 (pS8-Aβ) was purchased from Biopeptide (San Diego, CA, USA). Monomerised aliquots of peptides were prepared as described previously [15]. Before the experiment, a 2.5 mM peptide solution was prepared by adding 20 µL of 100% anhydrous DMSO (Sigma-Aldrich, St. Loius, MO, USA) to 0.22 mg of the peptide and was incubated for 1 h at room temperature, after which the peptide was diluted to the required concentration with a buffer solution or culture medium. In all experiments, only freshly prepared peptide solutions were used. As shown earlier, in such solutions, more than 80% of Aβ is in the monomeric form [16]. The concentration of diluted aliquots was routinely checked with BCA assay and was insignificantly different between the experiments and between the Aβ and pS8-Aβ preparations.

### 2.2. Cell Culture

Human HEK293 cells and mouse bEnd.3 cells obtained from ATCC were cultured in DMEM (Gibco, ThermoFisher Scientific, Waltham, MA, USA) containing 10% fetal bovine serum (FBS; Gibco, ThermoFisher Scientific, Waltham, MA, USA), 100 U/mL penicillin, 100 µg/mL streptomycin, sodium pyruvate and Glutamax (Gibco, ThermoFisher Scientific, Waltham, MA, USA) in T-25 and T-75 culture flasks at 37 °C and 5% CO_2_.

### 2.3. The Expression and Purification of Casein Kinase 2α (CK2α)

The plasmid was created on the basis of the bicistronic vector pET-21d-2c-5B [16600628] kindly provided by A.V. Ivanov (Engelhardt Institute of Molecular Biology). Rosetta (DE3) *E. coli* strains bearing plasmid pET-21d-2c-CK2α were grown overnight in 5 mL of LB medium supplemented with 10 g/L glucose, 150 mg/L ampicillin (A150), and 15 mg/L chloramphenicol (C15), at 37 °C. The overnight culture was harvested by centrifugation and the pellet was resuspended in 500 mL of fresh medium supplemented with A150 and C15. The cells were grown at 37 °C up to an optical density of 0.5 at 550 nm before isopropyl-D-thiogalactopyranoside was added to a final concentration of 1 mM. The cells were grown for additional 4 h and then harvested by centrifugation at 4000× *g* at 4 °C for 15 min. The cell pellet was resuspended on ice in 20 mL of buffer A (50 mM Tris–HCl, pH 7.0, 500 mM NaCl, 10% (*v/v*) glycerol, 5 mM 2-mercaptoethanol) supplemented with 1% (*v/v*) Triton X-100, protease inhibitors phenylmethylsulfonyl fluoride (PMSF, 1 mM), and 0.1% (*v/v*) of protease inhibitor cocktail (Sigma). The suspension was lysed by sonication on ice and pelleted at 10,000× *g* for 30 min. The clarified lysate was applied onto a 1-mL Ni-NTA-agarose column. The column was successively washed with buffer A (30 mL), buffer A containing 10 mM imidazole (30 mL), and then the protein was eluted by the same buffer containing 200 mM imidazole. Fractions (0.5 mL each) were analysed by SDS-PAGE. The target fractions were pooled, dialysed against buffer B (50 mM Tris–HCl, pH 7.0, 500 mM NaCl, 50% (*v/v*) glycerol, 5 mM 2-mercaptoethanol), and stored at −20 °C.

### 2.4. Phosphorylation and Dephosphorylation of Aβ in Cell Lysates

Radioactively labelled 32P-γ-ATP without a carrier (4000 Ci/mmol) was provided by the Shemyakin-Ovchinnikov Institute of Bioorganic Chemistry. HEK293 and bEnd.3 cells were grown in T-25 culture flasks until confluent, then harvested with TryplE (12604013, ThermoFisher Scientific, Waltham, MA, USA) and lysed in RIPA buffer (25 mM Tris-HCl, pH 7.6, 150 mM NaCl, 1% Nonidet-P40, 0.1% SDS, 1% sodium deoxycholate) containing either protease inhibitors (11836145001, Roche, Basel, Switzerland) and PhosStop phosphatase (4906837001 Roche, Basel, Switzerland) or protease inhibitors alone, with shaking at +4 °C for 1 h. The cell lysate was clarified by centrifugation at +4 °C for 10 min at 16,000× *g* and kept on ice for no more than 60 min before the start of the experiment. Protein concentration in the clarified lysates was determined using a BCA assay kit (ThermoFisher Scientific, Waltham, MA, USA). Protein concentrations in the clarified lysate measured with a BCA kit (ThermoFisher Scientific, Waltham, MA, USA) were 1.3 ± 0.2 mg/mL for HEK293 and 0.4 ± 0.1 mg/mL for bEnd.3. Phosphorylation of Aβ was performed in a 30 µL reaction mixture containing 10 µL of the clarified lysate, with protease and phosphatase inhibitors, 20 µL of PBS, 15 µM Aβ (from the stock solution in DMSO), and 5 mM MgCl_2_, in the presence or absence of 10 µM CK2 inhibitor TBBz (Sigma-Aldrich, St. Loius, MO, USA). The concentration of TBBz was chosen with regard to its toxicity towards HEK293 (Appendix A) and its reported Ki of 0.5–1 µM. Dephosphorylation of Aβ was carried out in a 30 µL reaction mixture containing 10 µL of a clarified lysate, with protease and phosphatase inhibitors or with only protease inhibitors, 20 µL of PBS, 15 µM pS8-Aβ (from the DMSO stock solution), and 5 mM MgCl_2_, in the presence of or in the absence of 100 nM okadaic acid (Sigma-Aldrich, St. Loius, MO, USA). The enzymatic reaction was initiated by adding a mixture of ^32^P-γ-ATP and ATP to final concentrations of 100 nM and 100 µM, respectively. The reaction was carried out for 30 min at 37 °C. At the end of the incubation, samples were mixed with 30 µL of Tris-Tricine Sample Buffer (Bio-Rad, Hercules, CA, USA) containing 2.5% β-mercaptoethanol, heated at 85 °C for 7 min, and stored at −20 °C. 

### 2.5. Phosphorylation of Aβ by Cell Surface Kinases

HEK293 and bEnd.3 cells were grown in T-25 culture flasks in DMEM with 10% FBS until confluent. Immediately before the experiment, the media was removed, and the cells were washed with extracellular phosphorylation buffer (Tris 30 mM, EDTA 0.5 mM, K_2_HPO_4_ 5 mM, MgCl_2_ 5 mM, NaCl 60 mM, CH_3_COONa 10 mM, pH 7.3) [17], after which they were incubated in 1.3 mL of the same buffer containing 10 μM Aβ, 5 mM MgCl2, 100 μM ATP and 10 nM ^32^P-γ-ATP (4000 Ci/mmol) for 30 min in a CO2 incubator at 37 °C. An equivalent mixture of deoxyribonucleoside triphosphates (dATP, dTTP, dGTP, dCTP) was added to some of the samples to a final concentration of 1.0 mM, together with the phosphatase inhibitor PhosStop. After the end of the incubation, the medium from the cells was collected, supplemented with 15 µL of HALT Protease/Phosphatase inhibitor cocktail (ThermoFisher Scientific, Waltham, MA, USA), aliquoted, and frozen. Cells were lysed in 1 mL of RIPA buffer containing protease and phosphatase inhibitors for 1 h on ice with shaking. Protein concentrations in the clarified lysate were measured by the BCA assay kit (ThermoFisher Scientific, Waltham, MA, USA) −2.0 ± 0.3 mg/mL for HEK293 and 0.8 ± 0.2 mg/mL for bEnd.3.

### 2.6. Intracellular Phosphorylation of Exogenous Aβ

HEK293 and bEnd.3 cells were counted, seeded at 50,000 cells per well of 12-well plates in DMEM with 10% FBS, and grown until confluent. The cells were washed once with Tyrode buffer, and 400 μL of Tyrode buffer containing 10 μM Aβ were added to the wells. The plate was incubated in a CO_2_ incubator at 37 °C for 24 h. Then, the buffer was replaced with Tyrode buffer (400 µL) containing 200 µCi phosphoric acid (4000 Ci/mmol) per well and incubated for 3 h at +37 °C. Cells were washed three times with cold PBS and lysed in 250 μL per well of RIPA buffer containing protease and phosphatase inhibitors for 1 h on ice with shaking. The lysates were clarified by centrifugation for 10 min at 16,000× *g* at +4 °C. One hundred microliters of the clarified lysate were mixed 1:1 with Tris-Tricine Sample Buffer containing 2.5% beta-mercaptoethanol, heated at 85 °C for 7 min, and stored at −20 °C.

### 2.7. Immunoprecipitation of Samples

For the immunoprecipitation of samples, 1E4E11 antibodies against pS8-Aβ (Sigma-Aldrich, St. Loius, MO, USA) or 4G8 antibodies against epitope 17–24 of Aβ (BioLegend, CA, USA) were conjugated with Dynabeads M-280 Sheep Anti-mouse magnetic beads (ThermoFisher Scientific, MA, USA) at a ratio of 1 μg of antibodies per 15 μL of beads according to the manufacturer’s protocol. The samples were diluted with PBS/0.1%BSA/1 mM EDTA wash buffer, pH 7.3, and then antibody-conjugated magnetic beads and the phosphatase inhibitor PhosStop were added. Immunoprecipitation (IP) was carried out for one hour at room temperature, the beads were washed three times with wash buffer on a magnetic rack, and then eluted by heating in 15 μL Tris-Tricine Sample Buffer at 85 °C for 7 min. Samples were loaded on SDS-PAGE or frozen and stored at −20 °C. See Table 1 for the IP mix composition for each experiment.

### 2.8. Analysis of Phosphorylation and Dephosphorylation of Aβ Using Monoclonal Antibodies

Pre-heated samples mixed with Tris-Tricine Sample Buffer were separated on a 12% Tris-tricine polyacrylamide gel and semi-dry transferred to a nitrocellulose membrane with a pore size of 0.2 μm (Bio-Rad, Hercules, CA, USA). The membrane was blocked for one hour in Pierce Clear Milk Blocking Buffer (ThermoFisher Scientific, Waltham, MA, USA) and incubated with primary antibodies 1E4E11 (Sigma-Aldrich, St. Louis, MO, USA) or 6E10 (BioLegend, CA, USA) overnight at +4 °C. After incubation with anti-mouse HRP-conjugated secondary antibodies GAM (Hytest, Turku, Finland), the membranes were imaged on a Bio-Rad ChemiDoc MP instrument (Bio-Rad, Hercules, CA, USA) using Supersignal West Pico PLUS Chemiluminescent Substrate (ThermoFisher Scientific, Waltham, MA, USA). Quantitate analysis was performed with the Image Lab 6.0.1 software (Bio-Rad, Hercules, CA, USA).

### 2.9. Analysis of CK2α and β-Actin Expression in bEnd.3 and HEK293 Cell Lines

For the analysis of CK2α expression, fresh cell lysates were prepared as described above and separated on SDS-PAGE using Bio-Rad Mini-Protean 10% Precast Gels according to the manufacturer’s protocol. The gels were transferred to a nitrocellulose membrane with a pore size of 0.45 μm (Bio-Rad, Hercules, CA, USA) and blocked for one hour in 5% non-fat milk (Bio-Rad). The blocked membranes were incubated overnight at +4 °C with primary antibodies sc-12738 (Santa Cruz, Dallas, TX, USA). Incubation with secondary antibodies and imaging was carried out as described above. The expression of β-actin was analysed analogously using primary antibodies ab6276 (Abcam, Cambridge, UK). 

### 2.10. Analysis of the Degree of Aβ Phosphorylation by Autoradiography

Pre-heated samples mixed with Tris-Tricine Sample Buffer were separated on a 12% Tris-tricine polyacrylamide gel and semi-dry transferred to a nitrocellulose membrane with a pore size of 0.2 μm (Bio-Rad, Hercules, CA, USA) or a PVDF membrane (Bio-Rad, Hercules, CA, USA, 1620137). Alternatively, the gel was placed on paper (Whatman 3MM, 0.34 mm) and dried under vacuum at 80 °C. The membrane or paper was coated with polyethylene film, exposed on a radiosensitive screen (BAS-IP MS 2025 E, GE Healthcare, Chicago, IL, USA) for 1–24 h, and analysed on a Typhoon FLA 9000 scanner (GE Healthcare Bio-Sciences, Uppsala, Sweden). Quantification and ratios of radioactively labelled components were obtained using the Image Lab 6.0.1 software (Bio-Rad, Hercules, CA, USA).

### 2.11. Synthesis of ^125^I-Labeled Protein Standards

#### 2.11.1. Synthesis of ^125^I-Aβ

A reaction mixture with the activity of 0.1 MBq/µL and a total sodium iodide concentration of 200 μM was obtained by mixing 10 µL of phosphate buffer (100 mM, pH 7.2) containing nonradioactive sodium iodide (200 μM) with 5 µL of ^125^INa (1 MBq/µL, Khlopin Radium Institute, St. Petersburg, Russia). Then, 12.5 nmol of Aβ in 10 µL of phosphate buffer (100 mM, pH 7.2) was added. The reaction mixture was cooled to 0 °C, and 12.5 µL (1.13 mg/mL) of water solution of chloramine T was added. After 5 min incubation at 0 °C, the reaction was terminated with 50 µL of 0.31 mg/mL dithiothreitol (aq). The buffer was exchanged using a Microspin™ G-25 Column (GE Healthcare, Sigma-Aldrich, St. Louis, MI, USA) equilibrated with phosphate buffer (100 mM, pH 7.2) by centrifugation at 1000× *g* for 3 min.

#### 2.11.2. Synthesis of ^125^I-Bovine Serum Albumin

Bovine serum albumin (BSA) solution from BCA assay kit (ThermoFisher Scientific, Waltham, MA, USA) containing 10 nmol of the protein was lyophilised and redissolved in 10 µL of phosphate buffer (100 mM, pH 7.2) containing nonradioactive sodium iodide (200 µM) and 5 MBq of ^125^INa. The reaction and purification was carried out as for ^125^I-Aβ.

### 2.12. Statistical Data Processing

All experimental data are shown as mean values ± standard deviations of the mean (SD), with the number of independent experiments indicated in Figure legends. The statistical difference between experimental groups was analysed by Student’s *t*-test. Probability values (*p*) less than 0.05 were considered significant. Statistical analysis was performed using the GraphPad Prism 9.1.2 software (GraphPad Software Inc., San Diego, CA, USA).

## 3. Results

### 3.1. Amyloid Beta Is Phosphorylated in HEK293 Cell Lysate at Ser8 with the Involvement of Casein Kinase 2

Aβ phosphorylation by intracellular kinases was probed by incubation of synthetic Aβ with HEK293 cell lysates in the presence of ATP and ^32^P-γ-ATP. As shown using specific antibodies to pS8-Aβ (1E4E11) (Figure 1a) and the analysis of ^32^P inclusion in the corresponding band on SDS-PAGE (Figure 1b), Aβ was indeed phosphorylated by HEK293 lysate. The phosphorylation was observed only in lysates of cells incubated at 37 °C in the presence of Aβ, but not in the absence of lysate, or in samples incubated on ice (Figure 1a,b).

The region of Aβ peptide starting with Ser8 residue (Ser8-Gly9-Tyr10-Glu11) coincides with the consensus sequence of CK2 (Ser-X-X-Asp/Glu/pTyr/pTre) [18]. It was hypothesised that CK2 might be involved in the phosphorylation of Aβ by HEK293 cell lysates. First, the research team tested the possibility of Aβ phosphorylation by CK2 using a purified recombinant enzyme. Indeed, in the presence of CK2, Aβ phosphorylation at Ser8 residue was observed, which decreased in the presence of a specific inhibitor of CK2, TBBz (Figure 1c). Phosphorylation of Aβ at Ser8 residue with HEK293 cell lysate in the presence of TBBz was decreased by 35% (Figure 1d). Taken together, these data suggest the involvement of CK2 in Aβ phosphorylation at Ser8 residue in HEK293 cells.

### 3.2. Exogenous Beta-Amyloid Is Not Phosphorylated by HEK293 Cells after Internalisation but Is Phosphorylated by Surface Kinases of These Cells

Extracellular Aβ is able to enter cells through various mechanisms, and the process of its internalisation is important for Aβ transport through the blood–brain barrier, utilisation, and its intracellular effects [19,20,21]. However, the process of Aβ metabolism and modification after the internalisation is not understood well. To study the phosphorylation of internalised Aβ, HEK293 cells were cultured for 24 h in phosphate-free Tyrode buffer in the presence of Aβ. Then, ^32^P-phosphoric acid was added to cell media for three hours’ incubation. The analysis with specific antibodies did not detect any pS8-Aβ in IP-enriched cell lysates (Figure 2a). Autoradiography analysis of lysates showed the presence of phosphorylated proteins across a broad range of molecular weights; however, bands corresponding to the molecular weight of ^125^I-labeled Aβ were not detected (Figure 2b). The profile of low-molecular-weight radioactive proteins in Aβ-treated and non-treated cell lysates also did not differ (Figure 2b). When cells were treated with synthetic pS8-Aβ, it was detected in IP-enriched lysates (Figure 2a), showing that HEK293 are able to internalise Aβ, but do not phosphorylate Aβ in this experimental setting.

Despite the absence of intracellular phosphorylation, exogenous Aβ was modified by surface kinases of HEK293 cells. HEK293 cells were incubated in a medium containing Aβ and ^32^P-γ-ATP for half an hour, after which the medium was collected and enriched by immunoprecipitation with antibodies to pSer8-Aβ (1E4E11) or to the 17–24 Aβ epitope (4G8) that did not contain phosphorylation sites. After enrichment for pS8-Aβ, the band corresponding to the phosphorylated Aβ was observed both when stained with specific antibodies (Figure 2c) and on the autoradiographs (Figure 2d). If IP was performed with 4G8 antibodies, pS8-Aβ phosphorylation was not detected with specific antibodies, but the incorporation of ^32^P into the corresponding band was still observed (Figure 2d). This observation implies the presence of Aβ species phosphorylated at residues other than Ser8; hence, these species are not stained with 1E4E11, but the incorporation of 32P is visible. Phosphorylation of Aβ at Ser8 was detected only in the presence of a cocktail of phosphatase inhibitors (Figure 2d). Thus, HEK293 cells are able not only to phosphorylate but also to dephosphorylate Aβ.

### 3.3. PP1 and PP2A Phosphatases Participate in Dephosphorylation of pS8-Aβ by HEK293 Cells

To assess Aβ dephosphorylation, pS8-Aβ was incubated with HEK293 cell lysates in the absence of phosphatase inhibitors at 37 °C. Alternatively, the incubation was carried out in the presence of okadaic acid, which is a selective inhibitor of protein serine/threonine phosphatases PP1 and PP2A. Dephosphorylation was assessed by incubating the membranes after Western blotting with antibodies 6E10 (Figure 3a). It was found that 6E10 antibodies bound to the 7–14 epitope of Aβ, with a 250-fold higher affinity for the non-phosphorylated beta-amyloid peptide compared to pS8-Aβ (Figure 3b). In the samples without inhibitor, intense staining with 6E10 antibodies was observed, while in the presence of okadaic acid, the average intensity of the bands decreased by almost 90%. Dramatic suppression of Aβ dephosphorylation by HEK293 cell lysate in the presence of okadaic acid indicates the key role of PP1 and PP2A phosphatases in this process.

### 3.4. Phosphorylation and Dephosphorylation of Aβ by bEnd.3 Cells

The endothelium of the BBB controls the entry of Aβ from the blood into the brain and its exit from the brain tissues into the peripheral circulation. However, the modifications that Aβ undergoes when crossing the BBB are not studied sufficiently. The authors evaluated the ability of mouse brain endothelial cells bEnd.3 to modify Aβ with intracellular kinases in lysate, with surface kinases, or after internalisation of exogenous Aβ by intact cells.

The specific antibodies for pS8-Aβ detected no phosphorylation of Aβ by the bEnd.3 lysate. However, the inclusion of radioactive ^32^P in the band corresponding to Aβ showed phosphorylation of Aβ by the lysate of bEnd3 cells, although lower than for HEK293 cells (Figure 4a). After the enrichment for pS8-Aβ with IP, followed by autoradiography, the band corresponding to Aβ was preserved for HEK293, but was not detected in the case of bEnd.3 (Figure 4b). Such difference in Aβ phosphorylation in the lysates correlates with drastically different expressions of CK2α catalytic subunit in HEK293 and bEnd.3 cells (Figure 4c).

As with HEK293 cells, when live bEnd.3 cells were treated with exogenous Aβ, phosphorylated peptide was not detected in the cell lysate either when using specific 1E4E11 antibodies (Appendix A) or when the inclusion of radioactive phosphorus in the band corresponding to Aβ was measured. The internalization of Aβ by bEnd.3 cells was confirmed (Appendix A). Phosphorylation of Aβ by bEnd.3 surface kinases was also not observed (Appendix A).

Next, the ability of bEnd.3 cells lysate to dephosphorylate pS8-Aβ was assessed. As in HEK293 cells, phosphatases PP1 and PP2A play a major role in pS8-Aβ dephosphorylation in bEnd.3 cells (Figure 4d). To compare the ability of HEK293 and bEnd.3 cells to dephosphorylate Aβ, the research team determined the level of npAβ in samples incubated in the presence of (*npAβ_inh_*, ng) and the absence of phosphatase inhibitors (*npAβ_total_*, ng), as well as the amount of protein lysate in each sample (*Q_prot_*, µg). The ability to dephosphorylate Aβ at the Ser8 residue (*I_dephosph_*, ng/µg) was calculated using the following formula:$$I_{dephosph}=\frac{\left(npA\beta_{total}-npA\beta_{inh}\right)}{Q_{prot}}\;,\;\mathrm{ng}/\mu g$$

The *I_dephosph_* value for HEK293 was almost four times lower than for bEnd.3 (Figure 4e), which indicates a higher activity of phosphatases in brain endothelial cells and may be one of the reasons for the low Aβ phosphorylation signal in bEnd.3.

## 4. Discussion

Phosphorylation of proteins is one of the most important regulators of cell metabolism, and its disruption leads to the development of various pathologies [22]. Thus, hyperphosphorylation of the cytoskeletal protein tau is observed in AD due to the activation of GSK3 kinase [23,24], mutations in the insulin receptor kinase lead to the development of diabetes, and aberrant activation of tyrosine kinases is a common cause of cancer [25]. Phosphatases are another part of the system that controls protein phosphorylation, and a change in their activity also leads to morbid consequences: increased activity of cdc25 tyrosine phosphatase is associated with a poor prognosis in multiple cancers [26], and a mutation in the Lyp phosphatase gene is a common risk factor for a number of autoimmune diseases [27]. There is a lot of evidence that the activity of kinases and phosphatases may change with age, and this change may be the cause of various disorders associated with ageing [28,29,30,31]. Age-related brain function disorders can also be associated with impaired protein phosphorylation: in an ageing organism, the basal level of brain AMPK activity increases [32], CREB phosphorylation in the hippocampus changes [33], and the level of GSK3β in the cholinergic system increases [34]. Age is a major risk factor for AD; however, the mechanism linking ageing with the development of AD has not yet been established. It is possible that aberrant protein phosphorylation observed in AD [35] may be not only a consequence but also one of the causes of the disease. Thus, kinases and phosphatases are potential targets for AD therapy and/or prevention. According to the amyloid hypothesis, the pathogenic cascade of AD is initiated by the accumulation and oligomerisation of the beta-amyloid peptide. Beta-amyloid contains a number of residues potentially capable of phosphorylation—these are the residues Ser8, Tyr10, and Ser26. The properties of Aβ phosphorylated at the Tyr10 residue are practically unstudied, and the first evidence for its presence in vivo was obtained only recently [36]. A peptide modified at Ser8 and Ser26 residues was found in the brain tissues of AD model mice [8,37], and the properties of these peptides were studied in in vitro models [9,10]. Since Aβ is formed at the outer side of the cell membrane [38] or is secreted after the formation in endosomes [39], the possibility of Aβ phosphorylation was first assumed in the extracellular space [11]. However, it was found that pS8-Aβ accumulated inside the brain neurons of AD model mice prior to the development of pronounced cerebral amyloidosis [7]. What is primary—the internalisation of Aβ or its phosphorylation—remains unclear. It was found that Aβ was able to be phosphorylated at the Ser8 residue by fresh HEK293 cell lysate and, to a lesser extent, by bEnd.3 brain endothelial cell lysate. On the other hand, if cells were treated with exogenous Aβ, intracellular phosphorylation of the peptide was not observed. It is possible that internalised Aβ remains trapped in the endosomal compartment and cannot interact with cytoplasmic kinases. Such localisation of Aβ is typical for brain endothelial cells that transport it across the BBB. For other cell lines, for example, human neuroblastoma line SH-SY5Y, a high degree of cytoplasmic localisation of internalised Aβ was shown [40]. Thus, intracellular Aβ phosphorylation preceding its aggregation is not excluded. One of the candidate kinases for intracellular phosphorylation is CK2. The authors of this paper have shown that it is able to phosphorylate Aβ in vitro and is likely involved in its phosphorylation in the cell lysate. Weaker phosphorylation of Aβ by bEnd.3 cells’ lysate compared to HEK293 correlates with much lower expression levels of CK2 in these cells. It has been previously shown that CK2 levels and activity are reduced in AD [41,42]. In addition to intracellular CK2, there are secreted forms of CK2 that modify extracellular proteins [17,43], including the extracellular sites in APP [44]. Since Aβ phosphorylation at the Ser8 residue prevents the formation of metal-dependent oligomers, inhibition of ion transport, and reduces cerebral amyloidogenesis [10], a decrease in CK2 activity may contribute to the development of AD.

In this work, it was shown for the first time that pS8-Aβ was a substrate for protein phosphatases PP1 and PP2A, and they played a major role in Aβ dephosphorylation. For this, a dinoflagellate toxin okadaic acid, [45], which inhibits these enzymes with high specificity, was used. bEnd.3 cells were shown to have three times’ higher ability to dephosphorylate pS8-Aβ than HEK293 cells. As reported previously, vascular deposits of pS8-Aβ are observed in only 60% of cerebral amyloid angiopathy cases [46], while the deposits of the pyroglutamylated Aβ form, independent of phosphatase activity, are found in more than 90% of vessel-associated Aβ aggregates. Relatively weak intracellular phosphorylation, lack of Aβ surface phosphorylation relative to HEK293, and high activity of phosphatases in bEnd.3 brain endothelial cells may constitute an explanation for this phenomenon.

Thus, casein kinase 2 is involved in Aβ phosphorylation in HEK293 cells, while PP1 and PP2A phosphatases are key contributors to pS8-Aβ dephosphorylation in HEK293 and bEnd.3. Compared to HEK293, bEnd.3 cells are less capable of Aβ phosphorylation, but have a significantly more pronounced ability to dephosphorylate pS8-Aβ. The data obtained allow the proposing of casein kinase 2 as a new target for the regulation of pS8-Aβ levels in the body and indicate a possible role of the BBB endothelium in controlling the status of Aβ modification.

Limitations: (1) The number of cells for both bEnd.3 and HEK293 samples represents that on T-25 flask at confluence; however, HEK293 cells grow more densely. Thus, the weak phosphorylation of bEnd.3 in the lysate and the absence of detectable surface phosphorylation may be due to the lower number of cells per sample than in HEK29. (2) The lysate was not separated into membrane and cytosolic fractions. Though the total protein content of cell membrane is orders of magnitude lower than that of cytoplasm, membrane kinases may account for some Aβ phosphorylation detected in the cell lysate. (3) The TBBz and okadaic acid are usually perceived as specific inhibitors of CK2 and PP1/PP2A, respectively. However, several studies report that these compounds have additional targets. In the concentration used in the study, okadaic acid also inhibits PP4, PP5, and PP6 phosphatases [https://doi.org/10.2174/0929867325666180508095242, https://doi.org/10.1385/1-59745-267-X:23], and TBBz would also have other targets such as PIM1/2, PKD1 and DYRK1a kinases [https://doi.org/10.1042/BJ20080309, https://doi.org/10.1016/j.jprot.2012.09.017, https://doi.org/10.1074/mcp.M700559-MCP200]. Thus, the involvement of CK2 in Aβ phosphorylation and of PP1 and PP2A in the dephosphorylation of Aβ should be validated using complementary methods or alternative inhibitors, such as K64 and K66 for CK2 [https://doi.org/10.1042/BJ20080309] (all accessed on 30 June 2022).

## Figures and Tables

**Figure 1 life-13-00147-f001:**
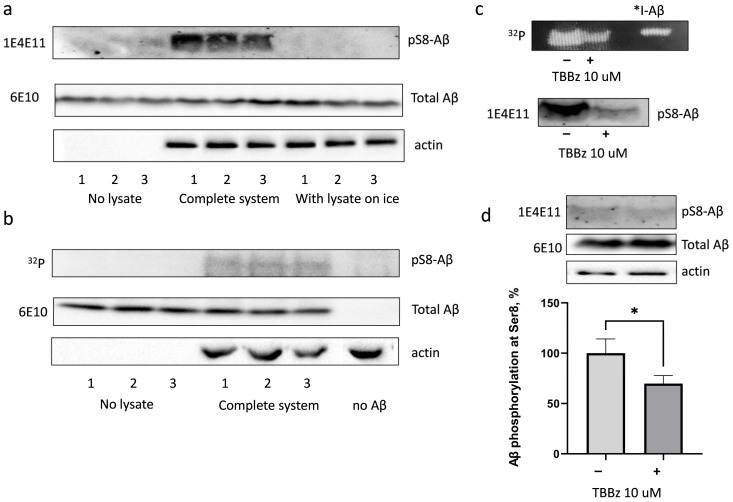
Phosphorylation of Aβ by HEK293 cells lysate (**a**,**b**,**d**) and recombinant CK2 (**c**) in the presence and in the absence of 10 μM TBBz, a CK2 inhibitor. Photographs of a nitrocellulose membrane after Western blotting, incubated with specific antibodies to pS8-Aβ (“1E4E11”), total Aβ (“6E10”) or actin, and an autoradiograph after SDS-PAGE (“^32^P”) are shown. I-Aβ—radioactive iodine-labelled Aβ standard. The Aβ phosphorylation at Ser8 residue by the HEK293 lysate was measured as the intensity of corresponding bands on nitrocellulose membranes stained with 1E4E11 antibodies normalized for actin. The results of three independent experiments measuring pS8-Aβ phosphorylation levels normalized to control (in the absence of CK2 inhibitor TBBz) are summarised in the graph (**d**). *—*p* < 0.05 by Student’s *t*-test.

**Figure 2 life-13-00147-f002:**
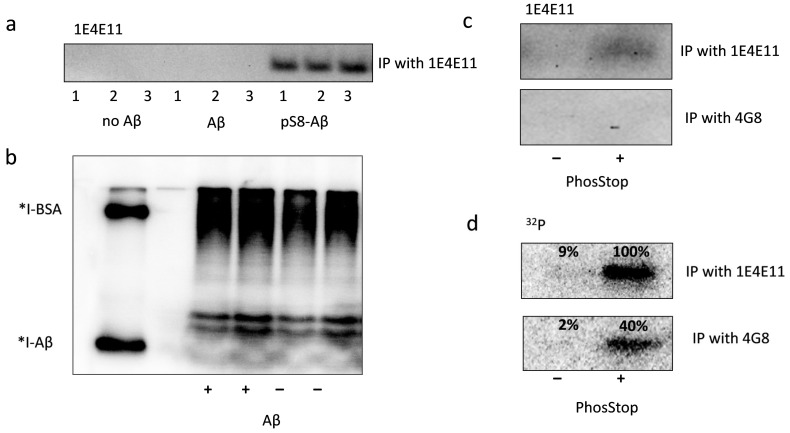
Phosphorylated Aβ in cell lysate (**a**,**b**) and in the extracellular medium (**c**,**d**) of HEK293 cells treated with exogenous Aβ. The extracellular medium and lysate were enriched for pS8-Aβ using IP with 1E4E11 antibodies (“IP with 1E4E11”) or for total Aβ using 4G8 antibodies (“IP with 4G8”). Photographs of nitrocellulose membranes after Western blotting, incubated with antibodies to pS8-Aβ (“1E4E11”) (**a**,**c**), and fragments of an autoradiograph after SDS-PAGE (“^32^P”) are shown (**b**,**d**). The numbers on the autoradiograph (**d**) indicate the relative intensity of the corresponding bands on WB. *I-Aβ, *I-BSA are radioiodine-labelled bovine serum albumin (BSA) and Aβ, respectively.

**Figure 3 life-13-00147-f003:**
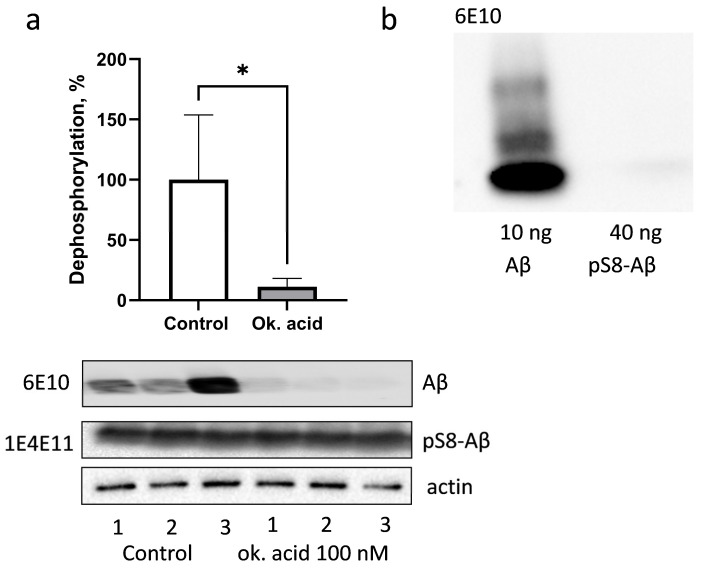
Dephosphorylation of pS8-Aβ in lysates of HEK293 cells (**a**). The result of three independent experiments is summarised on a graph (**a**), and photographs of representative nitrocellulose membranes after Western blotting, stained with 6E10 antibodies (**a**,**b**), 1E4E11 antibodies (**a**) or actin antibodies (**a**) are shown. The specificity of 6E10 antibodies to non-phosphorylated Aβ (**b**). A photograph of a nitrocellulose membrane after Western blotting is shown. *—*p* < 0.05.

**Figure 4 life-13-00147-f004:**
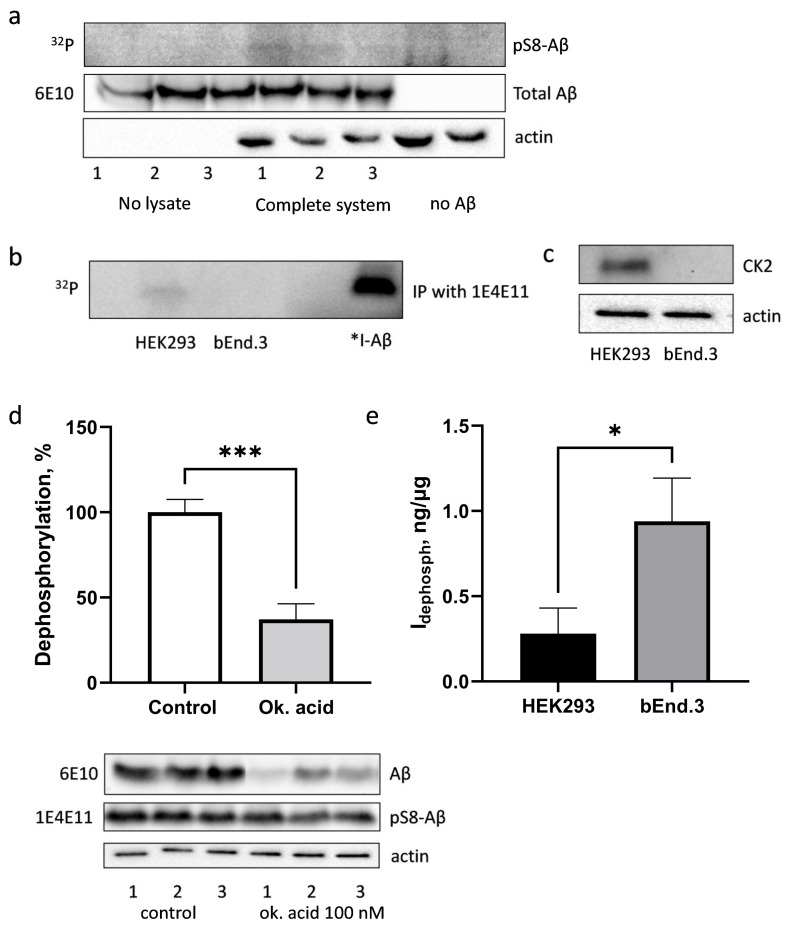
Phosphorylation of Aβ by bEnd.3 cells in the lysate (**a**,**b**). The level of CK2α protein in HEK293 and bEnd.3 (**c**). Dephosphorylation of pS8-Aβ by bEnd.3 cells (**d**) and by bEnd.3 and HEK293 cells in comparison (**e**). A fragment of an autoradiograph (“^32^P”) and photographs of nitrocellulose membranes after Western blotting stained with antibodies specific to pS8-Aβ (“1E4E11”), total Aβ (“6E10”), CK2α or actin are shown. *I-Aβ—radioactive iodine-labelled Aβ_42_ standard. Dephosphorylation of pS8-Aβ in lysates of bEnd.3 cells in the presence and absence of okadaic acid (**d**). The results of three independent experiments are summarised in histograms (**d**,**e**). *—*p* < 0.05, ***—*p* < 0.001.

**Table 1 life-13-00147-t001:** Composition of samples for immunoprecipitation.

Experiment	Sample Type	Sample, μL	IP Wash Buffer, μL	Magnetic Beads Suspension, μL	PhosStop, μL
Phosphorylation in cell lysates	Reaction mixture	30	470	20	50
Intracellular phosphorylation of exogenous Aβ	Cell lysate	120	450	10	50
Phosphorylation of Aβ by cell surface kinases	Cell medium	500	500	15	100

## Data Availability

Data is contained within the article or Appendix A.

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
