# Peer review of "Phosphorylation and Dephosphorylation of Beta-Amyloid Peptide in Model Cell Cultures: The Role of Cellular Protein Kinases and Phosphatases"

_life, 2023, doi:10.3390/life13010147_

Round 1

Reviewer 1 Report

The manuscript by Barykin et al. describes a set of experiment aimed to answer important questions about mechanisms of phosphorylation and dephosphorylation of amyloid beta peptide Aβ42. There is growing body of evidence that phosphorylation might be important modulator of the toxicity and aggregation propensity of this peptide but mechanisms regulating the level of its phosphorylated forms in different types of cells are not well understood. This study partially closed this gap in our knowledge and obtained results will be of interest for researchers and clinicians working in the field of Alzheimer’s and certain other neurodegenerative diseases. Experimental data are solid, generally well presented and support author’s conclusions. However, there are a few issues that should be addressed before the manuscript could be published.

Major:

1. In the experiments with bEnd.3 cells loaded with 32P phosphoric acid, the authors claimed that they did not detect Aβ phosphorylation. However, no blots/gels were presented to support this conclusion. Please, include it either in the manuscript or in the Supplementary.

2. For the same set of experiments, data confirming that exogenous Aβ entered bEnd.3 cells should be presented, similar to what authors demonstrated for HEK293.

3. On Fig. 2, B, showing data on HEK293 surface phosphorylation, there is no signal from 1E4E11 antibodies after IP with 4G8 (Fig. 2, B, top), though some incorporation of 32P into the peptide was detected (positive signal at the corresponding band in Fig 2, B, bottom). The authors should provide some discussion of this phenomenon.

Minor comments:

1. Throughout the MS, beta-amyloid peptide is sometimes referred to as “Aβ” and sometimes as “Aβ42”. Although I understood that no Aβ species of other lengths were used, it is important to be consistent and use the same term everywhere.

2. Methodology for 125I-BSA synthesis should be added to the Methods section

3. Labelling for some lanes on original blot image for Fig. 3, A, is missing and should be added

Reviewer 2 Report

 Immunochemical methods combined with
radioactive labelling were used by the authors to investigate in-vivo (in AD model mice cells) post translational modifications of amyloid beta (Aβ), namely its phosphorylation by
intracellular and surface protein kinases of HEK293 cells and brain
endothelial cells bEnd.3. The role of casein kinase 2 (CK2) and protein phosphatases PP1 and PP2A and involvement of blood brain barrier in the control of Aβ modification degree is demonstrated.  Aβ phosphorylation at Ser8 position was earlier detected in the brains of patients with AD and Down syndrome. The methods and results appear to be presented clearly and in adequate level of details, and the conclusions are supported by the reported results. Generally the design of experiments and presentation of results make the work easy to read and comprehend, however, minor English proofreading could further improve readability of the paper. E.g., definite article is missing before the second "modification" work in  line 51, the second comma in line 91 is unnecessary.  Comma in line 102 is unnecessary.  Definite article before "clarified lysates" is missing in line 134, both commas in lines 136-137 appear unnecessary, indefinite article before "clarified lysate" in line 139 should be replaced with definite. Line 144 - change "with protease inhibitors alone" to "only with protease inhibitors". Line 155 - delete "an" in "an extracellular phosphorilation buffer". Line 166-167 - Predicate is missing in "Protein concentrations  in the clarified lysate measured by the BCA assay kit", consider "were measured. Line 188-189 - -replaice "antibodies-conjugated" with "antibody -conjugated" or "magnetic beads, conjugated with antibodies".Line 227 -remove "a" in front of "water solution". Line 257 "The environment of the Ser8 residue in Aβ (Ser8-Gly9-Tyr10-Glu11) coin- 257 cides with the consensus sequence of SK2" - it is not clear from the context what is meant by "environment" here, consider using "domain" or something else less umbiguous. In line 427-429, minor rephrasing can improve style and readability , e.g. as in "bEnd3 was shown to have a three-times higher ability...".In line 437, changing  "... protein phosphotases PP1 and PP2A  are the key factor of PS8-Aβ phosphorylation" to "... PP1 and PP2A are key factors of (actors of, contributors to) PS8-Aβ dephosphorylation can be considered. Based on overall merit and quality of presentation of the paper, the reviewer recommends to accept the paper for publication after minor editorial revisions and additional round of proofreading. For information of the authors, MDPI has scheduled a special issue on AD and associated neurodegenerative diseases that might be of interest for them from the standpoint of contributing another research or review paper.  

Reviewer 3 Report

This manu submitted by Barykin et al. reports the phosphorylation and dephosphorylation of Aβ in two cell lines, HEK293 and bEnd.3. The authors conclude that an important role of CK2, PP1, and PP2A as regulators of Aβ phosphorylation and point to the involvement of the blood-brain barrier in the control of Aβ modification levels. However, the results do not well support the conclusions and the data is not appropriately presented. Extensive revisions are needed to improve its quality for publication.

1.      For all the figures, please modify the text font to make it uniform throughout the manuscript.

2.      When testing the phosphorylation of Aβ, for instance, in Figure 1A and B, can you show an extra line to show the total amount of Aβ (6E10 as the antibody) as a loading control? And I would suggest combining these results in A and B, and four groups of results (“No lysate”, “With lysate at 37C”, “With lysate on ice” and “no Aβ”) will be shown then.

3.      The upper image in Figure 1D is not clearly explained. How is the percentage of phosphorylation measured?    

4.      Figure 2 is poorly presented. It’s better to show the results in four panels (A,B,C,D). “IP with 4GB” is not explained in the legend.

5.      Line 299 and 302, there’s an error, Figure 2 is mistakenly described as “Figure 3”.

6.      Line 302, “Phosphorylation of Aβ…”, what is the evidence for this description?

7.      In figure 2B bottom, if 4GB has no phosphorylation site, why there is 32P signal?

8.      If just comparing Figure 3A top and Figure 4C without using the Idephosph, it seems that HEK293 cells perform higher dephosphorylation, which is confusing. I suggest defining and using Idephosph early in Figure 3.

Reviewer 4 Report

This study investigated the mechanism of regulation of Aβ phosphorylation, using immunochemical and cell biological techniques. The results indicated that CK2, PP1, and PP2 are involved in the regulation of Aβ phosphorylation. However, there are not enough experimental results to support the conclusion. To support the conclusions of this study, the following should be concerned.

1. The difference in kinetic activity or expression levels of casein kinase 2, PP1 and PP2 between HEK293 and b.End3 cells should be shown. The manuscript discuss the importance of the regulation of Aβ phosphorylation and the role of BBB in this cell type based on the results of different Aβ phosphorylation and dephosphorylation capacities in different cell types. The activity and expression levels of CK2, PP1, and PP2 in each cell type could support this conclusion.

2. Internal standards (ex. actin or other housekeeping proteins) should be shown in all Western blots. Internal standards are essential to discuss the signal intensity.

3. In Fig. 1C, the reason for setting the concentration of TBBz at 10 μM (is this the saturation level?) should be provided.

4. In relation to Fig. 1C and D, Do cell extracts in which CK-2 is removed with antibodies significantly reduce the phosphorylation level of Aβ?

5. Which aggregation stage (monomer, oligomer, protofibril, or fibril) of Aβ is most likely to be phosphorylated?

6. In the second paragraph of the subsection 3.2 (lines 292-304), Fig. 3B appears to be a mistake for Fig. 2B.

7. Regarding the autoradiography in Fig. 2A, the manuscript describes that Aβ is not phosphorylated because no signal of the same molecular weight as 125I-Aβ was obtained. However, the possibility that the aggregates may contain phosphorylated Aβ should be concerned. Have this study confirmed that there is no 32P signal in the sample immunoprecipitated with anti-Aβ antibodies such as 4G8 as in Fig. 2B?

8. As in the Western blot in Fig. 2B, what is the reason for the absence of signal with 4G8 but not with 1E4E11 antibody? Is it due to phosphorylation of other sites as described in the Discussion?

9. In relation to Q8, which site is phosphorylated should be identified not only by antibody but also by mass spectrometry.

10. Regarding Fig. 4B, is it possible to detect Aβ phosphorylation by b.End3 cell extracts even with phosphatase inhibitors such as PhosphoStop?

Round 2

Reviewer 3 Report

I'm satisfied with the response and my concerns are well addressed.

Author Response

On behalf of the Authors, I would like to express gratitude for the analysis of the manuscript, which helped to improve its quality and rigor.

Dr. Evgeny Barykin

Reviewer 4 Report

The revised manuscript responds well to the reviewer's comments. The difficulty in obtaining reagents and other materials for the experiments requested by the reviewer is acceptable as a practical problem. Considering this situation, the manuscript could be accepted if the followings are improved.

1. The most problematic aspect of this manuscript is that the evidence is too weak to conclude that CK2, PP1/PP2 are directly involved in Aβ phosphorylation in cells. Inhibitors can act on various kinases and phosphatases (refs. DOIs: 10.1042/BJ20080309, 10.1385/1-59745-267-X:23, 10.2174/0929867325666180508095242, 10.1074/jbc.M601054200, 10.1186/1471-2121-2-6, 10.1016/s0014-5793(98)00775-3). This is the reason why I proposed to experiment with cell extracts depreated with specific antibodies. On the other hand, it is understandable that antibodies are difficult to obtain. Therefore, I request that a description of the specificity of the inhibitors be added as a limitation of this study, including appropriate references. I also recommend that the specific statements of CK2 and PP1/PP2 after the colon in the title be changed to the general descriptions of kinase and phosphatase, respectively.

2. This study is largely based on Western blot analyses. In order to discuss the amount of target protein among different samples, it is essential to show the total protein level of each sample in the Western blot. Therefore, I requested that the internal standards be presented in the previous peer review. In the revised manuscript, internal standards were provided in most of the figures, but not enough in the followings.

Figure 2a

Supplementary Figures 2 and 3

In the next Western blot analysis, the total Aβ level should be indicated by CBB staining or silver staining, etc. Otherwise, the reader may have doubts about the presence of Aβ in the lane with no signal.

Figure 2c and d

Figure 3b

Supplementary Figures 2 and 4

3. The limitation (1) (lines 477-481) of the discussion section mentions a difference in the number of cells, which is contradicted by the fact that the internal standards in the corresponding experimental results in Figure 4c are at the same level.

4. In Figure 3, there is a (c) in the figure caption, but there is no c in the figure or in the main text.

Round 3

Reviewer 4 Report

The reviewers' concerns were clarified. This manuscript is ready for publication.